# Evaluation of Self-Management Behaviors and Its Correlation with the Metabolic Syndrome among the Type 2 Diabetes Mellitus Patients of Northern Saudi Arabia

**DOI:** 10.3390/jcm13010118

**Published:** 2023-12-25

**Authors:** Aseel Awad Alsaidan, Ashokkumar Thirunavukkarasu, Hassan H. Alhassan, Ibrahim Abdullah Bin Ahmed, Anas Salem Alnasiri, Wejdan Madallah Alhirsan, Nouf Nashmi M. Alazmi, Abdalaziz Khaled Alkhlaf, Jumanah Mohammed Alderbas, Motaz Abdulsalam Alkhaldi

**Affiliations:** 1Department of Family and Community Medicine, College of Medicine, Jouf University, Sakaka 72388, Saudi Arabia; ashokkumar@ju.edu.sa; 2Department of Clinical Laboratory Sciences, College of Applied Medical Sciences, Jouf University, Sakaka 72388, Saudi Arabia; h.alhasan@ju.edu.sa; 3Department of Family Medicine, College of Medicine, Imam Mohammad Ibn Saud Islamic University, Riyadh 11432, Saudi Arabia; dribrahima@hotmail.com; 4King Abdulaziz Specialist Hospital, Ministry of Health, Sakaka 72345, Saudi Arabia; aalnasiri@moh.gov.sa; 5College of Medicine, Jouf University, Sakaka 72388, Saudi Arabia; wejdan0116@gmail.com (W.M.A.); noufff5000@gmail.com (N.N.M.A.); azoz674@outlook.sa (A.K.A.); maazkm94@gmail.com (M.A.A.)

**Keywords:** metabolic syndrome, self-management, diet, exercise, lipid profile

## Abstract

Self-management behavior among diabetes patients is essential to monitor blood sugar levels, make necessary lifestyle changes, and help patients reduce their risk of complications from diabetes. We assessed the prevalence of metabolic syndrome (MS) and its association with self-management behavior and sociodemographic characteristics among 310 patients with type 2 diabetes mellitus (T2DM) attending different diabetes care centers in northern Saudi Arabia. We evaluated the self-management behaviors of patients with T2DM using a validated Arabic version of the Summary of Diabetes Self-Care Activities Scale. Regarding MS, we applied the National Cholesterol Education Program (NCEP) Adult Treatment Plan–3 (ATP–3) guidelines. A logistic regression analysis was used to identify the predictors of MS. We found that more than one-third (36.5%) of patients had MS according to the NCEP ATP-3 criteria. The prevalence of MS was significantly associated with unsatisfactory self-management behaviors. Regarding sociodemographic predictors for MS, we found a significant association between gender (ref: female: Adjusted OR (AOR) = 1.89, 95%CI = 1.17–2.95, *p* = 0.007) and body mass index (ref.: normal range: AOR = 2.98, 95%CI = 1.31–5.07, *p* = 0.003). Our findings suggest a tailor-made multifaceted intervention to improve the self-management behaviors of T2DM patients, which, in turn, can reduce MS.

## 1. Introduction

The prevalence of type 2 diabetes mellitus (T2DM) in recent years has become a significant global health problem. Research shows that T2DM is the most common and clinically crucial pathophysiologic disorder [1,2]. By 2045, projections from the International Diabetes Federation (IDF) indicate that one in every eight adults, about 783 million, will live with diabetes [3]. The prevalence of T2DM is rising rapidly not only in developed countries but also in developing countries like the Kingdom of Saudi Arabia (KSA) also observed an increasing prevalence [3,4]. This could be due to the aging population, obesity, and sedentary lifestyle [5,6]. For many years, T2DM was considered an adult disease, but with the rising teenage and child obesity cases, this phenomenon has changed [7,8].

Managing this major public health problem necessitates a multidimensional approach that includes self-care behaviors [9]. Self-management behavior is essential for people with diabetes because it helps them monitor their blood sugar levels and change their diet and exercise habits if necessary [9]. Hence, their quality of life can be significantly improved [10]. Self-management behavior can also assist patients to reduce their risk of complications from diabetes. Several types of self-management behavior, including diet, exercise, medication management, and communication, are essential in the care process for type 2 diabetic patients [9,11,12]. Diet is one of the most critical types of self-management behavior for people with diabetes [9,13]. People with diabetes should follow a healthy diet with plenty of fruits and vegetables, whole grains, low-fat dairy products, and lean protein sources. They should also avoid sugar-sweetened beverages, processed foods, and excessive amounts of salt [14,15]. Exercise is also crucial for people with diabetes. They should aim for at least 30 min of physical activity per day. Moderate-intensity exercise (such as brisk walking) is best for diabetes patients, while vigorous-intensity exercise (such as running) may be too strenuous. Medication management is another crucial type of self-management behavior for people with diabetes [16].

As the prevalence of T2DM increases, so does the incidence of metabolic syndrome (MS). MS is a group of conditions that increase the risk of developing stroke and cardiovascular events [17,18]. A study conducted by Ji M et al. in China stated that self-management behaviors, especially physical activity, are one of the critical predictors that were associated with MS among patients with diabetes in their study [19]. In evaluating the self-management behaviors among Saudi T2DM patients by Alodhayani A et al. in 2021, they revealed that gender and marital status were the significant associated factors [20]. Another study from Jordan among T2DM patients stated that nearly three-fourths of their participants had unsatisfactory self-management behavior [21]. The prevalence of MS among Saudi diabetic patients is relatively high, which increases the chance of serious health complications occurring [22]. However, there is limited research on self-management behavior and correlation with MS among T2DM patients in Saudi Arabia, especially in the Aljouf region of KSA. This means we do not know how well it works or what its effects are on people’s quality of life, so we need to do more research in this area.

Furthermore, an ongoing assessment of the epidemiological data related to self-management is essential, as these behaviors are constantly changing. Additionally, this research could contribute to the existing literature and to the understanding of healthcare professionals of how self-management behavior can improve people’s diabetes outcomes. Therefore, the present study was aimed to assess the self-management behaviors and their correlations with MS and other sociodemographic characteristics of the T2DM patients attending outpatient diabetes care clinics.

## 2. Participants and Methods

### 2.1. Study Design

This study is a quantitative cross-sectional study that was conducted from May 2023 to November 2023.

### 2.2. Study Setting

The present study’s patients were selected from those attending outpatient diabetes care clinics in the Aljouf region of KSA. This region is located close to Jordan in the northern part of KSA. The present study included adult T2DM patients (18 years and older) attending outpatient diabetes care clinics. We excluded the other forms of diabetes, inpatients, newly diagnosed patients, and those unwilling to participate.

### 2.3. Sampling Method

We have used the open-source, freely available Raosoft online sample size calculator to find the minimum number of T2DM patients required for this survey [23]. The Raosoft calculator uses the same sampling strategies as the WHO and Cochran sample size estimation formula (n = z^2^pq/e^2^). The research team considered anticipated self-management behavior based on a study conducted by Almomani MH et al. (*p* = 20.8%) [21], an acceptable margin of error (e = 5%), and 80% power to estimate the sample size. However, the research team included an additional 20% of the sample size. Hence, the calculated minimum required sample size was 300.

### 2.4. Data Collection Steps

Firstly, we received the necessary ethical clearance and approvals from the concerned institutions to start the data collection. Next, after each patient’s follow-up at the outpatient clinic, we briefed the study objectives to the participants and obtained informed consent in accordance with the declaration of Helsinki. Next, we asked them to fill out a standard, a pretested Arabic version of the data collection proforma consisting of three sections. The collected data were anonymous and de-identified. The first section asked about the background of diabetes ’patients and health-related characteristics. After that, the data collectors recorded the following details in the second section from patients’ files: blood pressure level (systolic/diastolic) (mmHg), height (cm), weight (kgs), waist circumference (cms), and laboratory tests (fasting blood sugar and lipid profile). The included test results were from the latest visit laboratory results. We followed the National Cholesterol Education Program Adult Treatment Plan–3 (NCEP ATP–3) criteria as an operational definition to classify the presence or absence of MS among the participants. The NCEP ATP–3 criteria are one of the widely used criteria in clinical and epidemiological studies to assess MS [24]. Please find the criteria and values in Appendix A. We calculated the body mass index (BMI) using the standard formula [BMI = weight (kgs)/height (m^2^)]. Furthermore, we classified the BMI status of the patients according to the World Health Organization (WHO) classification [25]. Similarly, we followed the standard protocols to measure the waist circumference of the patients [26,27].

The third part of the questionnaire inquired about the diabetes patients’ self-care behavior. The Summary of Diabetes Self-Care Activities (SDSCA-Arabic), a widely used tool to assess self-care behaviors for diabetic patients, was used in the present study [21,28,29]. The questionnaire comprised five significant sectors to access following self-care for diabetes patients. Firstly, the diet sector includes general and specific diets containing two and three items, respectively. The blood glucose testing, medication, and exercise sectors are all made up of two things each. The remaining part of the questionnaire is the foot care sector, which consists of four components. Hence, a total of 15 questions were used to evaluate the self-management behavior of the participants. An eight-point Likert scale was used for the past week’s assessment of self-care performance frequency. The minimum frequency is zero days, while the maximum is seven days. In the diet sub-scale, high-fat consumption is assessed on item five. It is coded in reverse for accurate recording to display adherence to a low-fat diet for the diabetic patient. For each subscale, the mean score is calculated following the guidelines provided by SDSCA. Afterward, the overall average of the mean scores of the subscales is obtained. This average of the total score also ranged from zero to seven.

Different scores reflected different levels of self-care for diabetic patients. For example, higher scores reflect higher self-care than lower scores. We categorized the scores ranging from zero to four as unsatisfactory, whereas those ranging from five to seven were considered satisfactory. The SDSCA-Arabic has, over time, been proven to be reliable and valid. Furthermore, we pretested the Arabic version of SDSCA through the pilot study among 30 T2DM patients, and the Cronbach alpha value for the tool was 0.79.

### 2.5. Statistical Analysis

The survey team used Statistical Package for the Social Sciences (IBM SPSS Statistics for Windows, version 21.0, Armonk, NY, USA: IBM Corp.) for data curation and necessary statistical tests. We reported the descriptive data as numbers, proportion (%) for discrete data, and mean with standard deviation for the quantitative data. We executed Spearman’s correlation test (test value is presented as rho) to identify the correlation between the scores of each SDSCA item and the MS components. The research team performed a multivariate analysis (binomial logistic regression) to find the association between MS, self-management behaviors, and sociodemographic characteristics. A *p*-value of less than 0.05 was identified through binomial logistic regression and was considered a statistically significant variable.

## 3. Results

During the data collection period, we communicated with 353 T2DM patients, and 310 patients (more than minimum required) agreed to participate in the study (response rate = 87.8%). Of the patients who participated in the study, the majority were between 50 and 60 years old (40.6%), male (52.6%), married (75.8%), and non-smokers (79.7%). The mean ± SD of the duration of diabetes among the participants was 9.84 ± 5.23, and most patients were only on oral hypoglycemic agents (OHA) (Table 1).

Values of physical parameters, such as BMI (kg/m^2^), waist circumference (cm), systolic and diastolic BP (mmHg), and laboratory values of participants, are presented in Table 2.

Of the 310 participants, 34 patients BMI were within the normal range (11%), 110 (35.5%) were overweight, and 166 were obese (53.5%) (Figure 1).

Table 3 shows the mean and SD scores of each subscale and the overall scores of SDSCA. Our study observed the lowest scores for participating in a specific exercise session (mean = 1.84) and the highest scores for insulin injections according to the recommended schedule (mean = 5.93). After applying the set criteria (overall mean score < 4 is considered unsatisfactory, and from 4 to 7 is considered satisfactory), we found that about 69% of the patients’ self-care behaviors were unsatisfactory.

The Spearman correlation test revealed that the number of days following a healthy diet plan is negatively correlated with waist circumference (cms) (rho = −0.252 and *p* = 0.001) and fasting blood sugar (mmol/L) (rho = −0.260 and *p* = 0.001). In contrast, HDL level (mmol/L) was positively correlated with the healthy diet plan (rho = 0.168 and *p* = 0.003). Regarding physical activity, we found a positive correlation with HDL (mmol/L) (rho = 0.139 and *p* = 0.014) and negative correlation with the BMI (kg/m^2^) (rho = −0.206 and *p* = 0.001). On adhering to the treatment regimen, fasting blood sugar (mmol/L) was significantly and negatively associated with both insulin (rho = −0.311 and *p* = 0.001) and OHA (rho = −0.250 and *p* = 0.006) adherence (Table 4).

After applying the NCEP ATP 3 criteria, we found that 36.5% of the patients in the present study were considered to have MS (Figure 2).

After adjusting other variables in binary logistic regression, the presence of MS among the participants was significantly associated with gender (ref: Males: adjusted OR [AOR] = 1.89, 95%CI = 1.17–2.95, *p* = 0.007), smoking status (ref: Non-smokers: AOR = 1.56, 95% CI = 1.56 (1.17–2.41, *p* = 0.016), self-management behavior (ref.: satisfactory: AOR = 1.55, 95%CI = 1.11–2.87, *p* = 0.027), body mass index (ref.: normal range: AOR = 2.98, 95%CI = 1.31–5.07, *p* = 0.003), and the presence of comorbid conditions (ref: No: AOR = 1.74, 95% CI = 1.11–2.67, *p* = 0.006) (Table 5).

## 4. Discussion

Globally, the significance of a holistic and multidimensional approach is proven, not only when it comes to T2DM but also to the cluster of comorbidities that may be associated, including MS [17]. The present study evaluated self-management behavior and its association with MS and sociodemographic factors among the T2DM patients.

Regarding self-management behavior, the present study found the lowest scores in participating in a specific exercise session and the highest scores in taking insulin injections according to the recommended schedule. Interestingly, a study conducted in a tertiary care center in the KSA and another study by Almomani et al. also found that the lowest score was in exercise sessions and the highest score on the medication adherence scale; however, their scores were slightly higher than the present study in most of the subscales [21,30]. Even though a community-based survey conducted in India among diabetes patients showed similar trend findings across the subscales of self-management behavior, their participants’ overall scores were lower than our study [31]. We identified the major area of concern in self-care management, which is regular exercise, an essential component of self-care behavior that is to be followed. In addition to routine diabetes care, healthcare providers must consider giving special attention to improving regular exercise among T2DM patients, as exercise is proven to reduce the cluster of conditions of MS [32,33]. Our study findings were similar to the studies mentioned above. We found that those who followed more days of physical activities had low BMI, low TGL, and high HDL.

One of the positive and encouraging findings identified in the present study was that the highest mean score was observed in the number of days adhering to insulin injection (mean = 5.93), followed by OHA intake (mean 5.72). Furthermore, adhering to the treatment regimen for more days negatively correlates with fasting blood sugar (less). These findings indirectly indicate the medication adherence practice among T2DM patients. The relatively good adherence suggests the impact of previous studies and incorporating their suggestions into the diabetes care plan [34,35,36]. According to the overall score’s categorization, over two-thirds of the study participants had unsatisfactory self-management behaviors. This finding is similar to several studies conducted among diabetes patients, indicating that self-management behaviors need to be specially addressed by policymakers [21,37,38,39].

The present study revealed that 36.5% of the T2DM patients identified having MS, a clinically significant finding for healthcare providers and policymakers. Previous researchers reported wide variations in the prevalence of MS [17,40,41,42]. These variations are due to operational definitions of MS, healthcare settings of the included patients, and availability of healthcare services. These findings from the present study and other studies are noteworthy as they underscore the high prevalence of MS among diabetes patients. Furthermore, these revealed findings insist on the interconnection of diabetes and MS, which share common risk factors and pathological pathways. The presence of these disorders together can have a synergistic effect, increasing the likelihood of developing more complications of T2DM [43,44]. Hence, clinicians and policymakers consider implementing regular screening to identify the cluster of risk factors related to MS to prevent or delay complications.

We found that MS among the T2DM patients was significantly associated with self-management behavior. Similar to the present finding, Ji M et al. in China found a significant positive correlation between MS and self-management behavior, especially the physical activity subscale [19]. It is worth mentioning here again that the exercise subscale mean score of the present study was the lowest among all subscales. One of the most likely reasons might be the bi-directional association between self-management and MS. Numerous pieces of evidence suggest that patients who were identified with MS tend to have a higher risk of developing depression and a lack of support for social activities that help them to implement self-management behaviors such as exercise and diet [45,46]. Simultaneously, poor diabetic self-management behaviors may lead to the development or worsening of MS [46,47].

Other critically associated factors of MS that were identified through logistic regression analysis were gender, smoking, and BMI. Considering the conservative nature of Saudi society and sociocultural reasons, the gender differences in the prevalence of MS are a noteworthy finding. Some authors have found a similar association, while some other studies varied in their conclusions [19,48,49]. The differences might be due to sociocultural settings and the presence of other comorbidities. Interestingly, a study performed by Du S et al. found that MS was predominantly higher among males than females [50]. Nonetheless, the influence of gender-specific variation in the prevalence of MS indicates tailor-made, gender-specific targeted intervention strategies are required to reduce the MS burden among T2DM patients. Another important finding of the present study was smokers had higher odds of developing MS among T2DM patients than non-smokers, indicating the role of smoking on the development of MS. Hence, smoking is a non-modifiable risk factor, regular targeted health education and smoking cessation sessions are to be taken to the patients. The strong association between BMI and MS in the present study is consistent with the existing literature, suggesting that weight management programs for diabetes are to be implemented as a part of routine diabetes care [51,52]. For instance, Kobo O et al. stated that BMI is a significant factor with a strong negative predictive value, and the presence of normal BMI is one of the highly supporting pieces of evidence to rule out MS [53]. Healthcare providers and policymakers can take the practical values of these findings in the following ways. Firstly, the risk factors can be used as surrogate markers to identify the T2DM patients who are at risk of presenting MS. Hence, tailored and early intervention strategies can be implemented through the multidisciplinary approach. Furthermore, through health education programs using a multidisciplinary approach, the healthcare providers should support the patients’ empowerment, as self-management behavior among T2DM patients can be attained primarily through patient empowerment [9,54,55,56].

The research team made the best possible attempt with a standard method for this study. The present study provides valuable insights into one of the important and rising health problems. We used a standard and validated tool to evaluate the self-management behavior of T2DM that can help the policymakers to implement necessary measures. Nonetheless, the following constraints are to be noted while interpreting this paper. Firstly, the present study recruited T2DM patients from outpatient diabetes care centers; hence, the findings cannot be the same for the inpatients and other forms of diabetes. Next, considering the nature of the cross-sectional study design, we could not evaluate the causal association between MS and self-management behaviors. Finally, considering the wide range of differences in socio-economic and cultural settings in the KSA, this study’s findings may not be applicable to other regions.

## 5. Conclusions

The present study observed that a high proportion of T2DM patients identified with MS. The self-management behaviors towards diabetes care were unsatisfactory among more than two-thirds of patients. Prevalence of MS is significantly associated with unsatisfactory self-management behaviors, gender, smokers, and BMI status. Our study findings suggest that tailor-made targeted measures aimed at improving the self-management behaviors of T2DM patients. Thus, implementing adequate self-management behaviors can help in achieving optimal clinical outcomes in patients with T2DM. Hence, the patients can effectively participate in their care through diet and physical activity modifications, adhering to treatment plans, and periodic health monitoring. Furthermore, we recommend conducting prospective exploratory studies across different regions of KSA to recognize the region-specific intervention measures and temporal association.

## Figures and Tables

**Figure 1 jcm-13-00118-f001:**
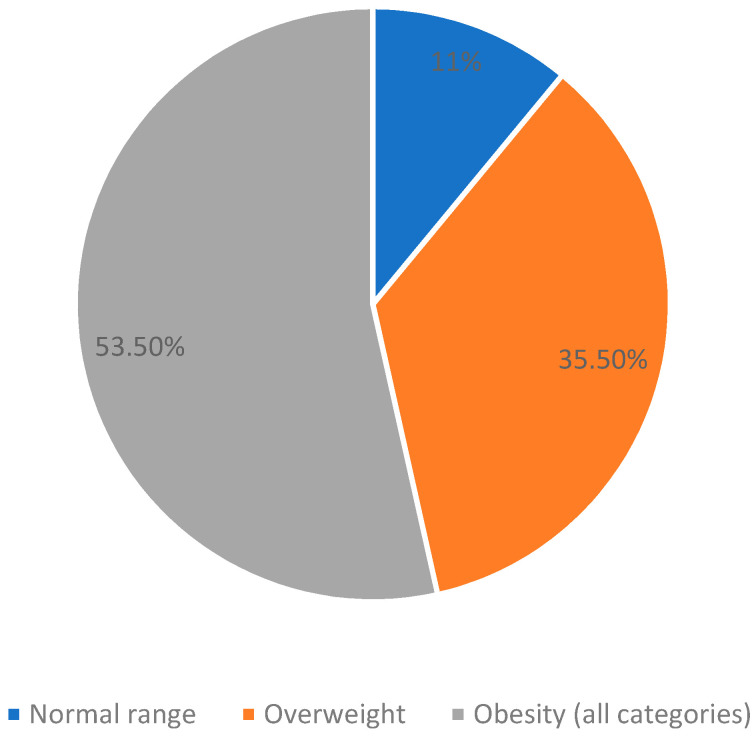
BMI categories (n = 310).

**Figure 2 jcm-13-00118-f002:**
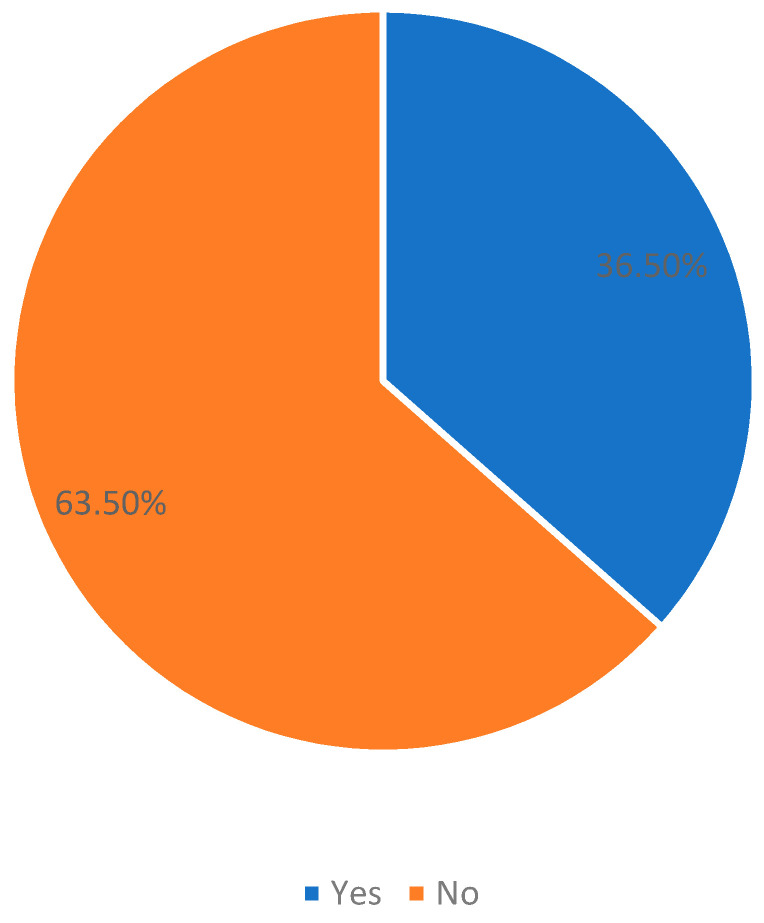
Prevalence of metabolic syndrome (MS) among T2DM patients of the present survey (n = 310).

**Table 1 jcm-13-00118-t001:** Sociodemographic characteristics of the participated type 2 diabetes mellitus (T2DM) patients (n = 310).

Variables	Frequency (n)	Proportion (%)
Age (mean ± SD)	55.50 ± 11.06	
Less than 50 years	91	29.4
From 50 to 60 years	126	40.6
More than 61 years	93	30.0
Gender		
Male	163	52.6
Female	147	47.4
Educational qualification		
No formal education.	46	14.8
Primary/Preparatory/Secondary	173	55.8
University/Graduate Studies	91	29.4
Occupation		
Employee in a government sector	87	28.1
Employee in the private sector	18	5.8
Unemployed	111	35.8
Retired	94	30.3
Married status		
Single	75	24.2
Currently married	235	75.8
Monthly income in SAR (1 USD = 3.75 SAR)		
Less than 5000 SAR	100	32.2
From 5000 to 1000 SAR	104	33.5
More than 10,000 SAR	106	34.3
Smoking status		
No	247	79.7
Yes	63	20.3
Duration of T2DM (mean ± SD)	9.84 ± 5.23	
Type (s) of anti-diabetes treatment		
Insulin only	82	26.5
Oral hypoglycemic agents (OHA) only	204	65.8
Both	24	7.7
Presence of comorbid condition		
No	208	67.1
Yes	102	32.9

**Table 2 jcm-13-00118-t002:** Physical and laboratory measurement of the T2DM patients (n = 310).

Parameters	Mean	SD
BMI (kg/m^2^)	32.02	13.06
Waist circumference (cms)	105.75	16.22
Systolic BP (mmHg)	132.95	14.41
Diastolic BP (mmHg)	79.88	9.65
FBS (mmol/L)	8.56	6.96
LDL ((mmol/L)	2.85	0.93
HDL (mmol/L)	1.24	0.68
Total Cholesterol (mmol/L)	4.65	1.09
TGL (mmol/L)	1.68	0.78

**Table 3 jcm-13-00118-t003:** Self-management behavior patterns among the participants evaluated by the Arabic version of Summary of Diabetes Self-Care Activities (SDSCA) (n = 310).

Subscales/Items	Mean	SD
Diet (general and special)		
1. Followed a healthy eating plan last week	2.80	2.59
2. On average, the number of days followed the diet plan in a week	2.81	2.62
3. Eat five or more servings of fruits and vegetables in a week	3.67	2.45
4. Evenly distribution of carbohydrates in food	3.65	2.52
5. Eat high-fat foods such as red meat or whole-fat dairy products	3.83	2.30
Physical activities		
6. Involved in at least 30 min of physical activity (continuous such as walking)	2.65	2.64
7. Participate in a specific exercise session (such as swimming, cycling)	1.84	2.37
Blood sugar testing		
8. Testing for blood sugar	3.78	2.76
9. Number of times testing blood sugar according to the recommendation by the doctor	3.20	2.88
Footcare (how many days in a week)		
10. Checked the feet	1.66	2.55
11. Inspect the inside of your shoes	1.42	2.47
12. Washed with the soap	4.99	2.92
13. Dry between your toes after washing	2.82	2.97
Medication		
14. Taking insulin injections according to the schedule	5.93	2.05
15. Taking OHA according to the recommendation	5.72	2.17
Overall (mean ± SD)	3.98	1.26
Unsatisfactory (<4) = 68.39%		
Satisfactory (from 4 to 7) = 31.61%		

**Table 4 jcm-13-00118-t004:** Relationship between each component of SDSCA and BMI values, lipid profile, and fasting blood sugar: The data presented here are Spearman’s rho/*p*-value (*p* <0.05 is a statistically significant variable).

SDSCA Items	BMI (kg/m^2^)	Waist Circumference (cms)	Total Cholesterol (mmol/L)	Triglycerides (mmol/L)	LDL (mmol/L)	HDL (mmol/L)	Fasting Blood Sugar (mmol/L)
Healthy diet plan	−0.160/0.005	−0.252/0.001	−0.071/0.212	−0.172/0.002	−0.069/0.227	0.168/0.003	−0.260/0.001
Specific diet plan	−0.125/0.027	−0.196/0.001	−0.035/0.545	−0.172/0.002	−0.008/0.882	0.176/0.002	−0.248/0.001
Fruits and vegetables intake	0.063/0.266	−0.012/0.839	−0.013/0.816	−0.077/0.175	−0.038/0.507	0.107/0.61	−0.133/0.019
Evenly distribution of carbohydrates in food	0.037/0.521	−0.013/0.824	−0.048/0.396	−0.115/0.044	−0.044/0.441	0.103/0.069	−0.151/0.008
High fat intake	0.001/0.996	0.013/0.813	0.038/0.502	0.169/0.029	0.180/0.013	−0.004/0.943	0.021/0.712
Physical activity	−0.206/0.001	−0.229/0.001	−0.031/0.586	−0.075/0.188	0.019/0.734	0.139/0.014	−0.152/0.008
Specific exercise session	−0.236/0.001	−0.230/0.001	0.032/0.574	−0.137/0.016	0.050/0.378	0.159/0.005	−0.199/0.001
Testing for blood sugar	0.064/0.261	0.009/0.877	−0.009/0.870	−0.010/0.858	−0.035/0.541	0.124/0.029	0.031/0.582
Number of times testing blood sugar	0.051/0.374	0.017/0.772	−0.062/0.278	0.030/0.605	−0.100/0.077	0.100/0.079	0.001/0.984
Checked the feet	0.019/0.739	−0.048/0.399	−0.041/0.476	−0.196/0.001	−0.018/0.752	0.021/0.710	0.007/0.905
Inspect the inside of your shoes	−0.002/0.967	0.025/0.662	−0.075/0.187	−0.209/0.001	−0.059/0.297	0.093/0.102	−0.033/0.562
Washed with the soap	0.005/0.930	0.025/0.667	−0.145/0.011	0.013/0.817	−0.164/0.004	0.035/0.544	0.045/0.425
Dry between your toes after washing	0.026/0.644	−0.015/0.788	−0.145/0.426	−0.012/0.835	−0.031/0.592	0.052/0.365	−0.092/0.106
Adherence to insulin	0.073/0.198	0.002/0.973	−0.143/0.012	0.068/0.233	−0.209/0.001	0.080/0.161	−0.311/0.001
Adherence to OHA	0.161/0.004	0.117/0.040	−0.134/0.018	0.113/0.047	−0.177/0.002	0.071/0.215	−0.250/0.006

**Table 5 jcm-13-00118-t005:** Self-management behaviors and other factors associated with MS. Test applied: multivariable analysis (Binomial logistic regression).

Characteristics	Total Participants (n = 310)	MS		Univariate Analysis No vs. Yes		Regression Analysis * No vs. Yes	
		No (%) n = 197	Yes (%) n = 113	Unadjusted Odds Ratio (OR) (95% CI)	*p* Value **	Adjusted OR (AOR) (95% CI)	*p* Value **
Age							
Less than 50 years	91	56	35	Ref		Ref	
From 50 to 60 years	126	85	41	0.77 (0.44–1.36)	0.367	0.79 (0.40–1.52)	0.482
More than 61 years	93	56	37	1.06 (0.59–1.91)	0.854	1.04 (0.46–2.35)	0.929
Gender							
Male	163	110	53	Ref		Ref	
Female	147	87	60	2.37 (1.50–3.18)	0.026	1.89 (1.17–2.95)	0.007
Educational Qualification							
No formal education.	46	27	19	Ref		Ref	
Primary/Preparatory/Secondary	173	106	67	0.87 (0.46–1.74)	0.375	0.89 (0.41–1.92)	0.766
University/Graduate Studies	91	64	27	0.60 (0.29–1.26)	0.839	0.63 (0.24–1.71)	0.365
Occupation							
Government sector	87	59	28	Ref		Ref	
Private sector	18	7	11	3.31 (1.16–5.45)	0.025	2.75 (0.90–5.23)	0.075
Unemployed	111	68	43	1.33 (0.74–2.40)	0.34	1.12 (0.43–2.89)	0.815
Retired	94	63	31	1.04 (0.56–1.93)	0.909	0.96 (0.43–2.15)	0.921
Married status							
Single	75	42	33	Ref		Ref	
Currently married	235	155	80	0.66 (0.39–1.12)	0.12	0.71 (0.39–1.28)	0.229
Monthly income in SAR (1 USD = 3.75 SAR)							
Less than 5000 SAR	100	64	36	Ref		Ref	
From 5000 to 1000 SAR	104	60	44	1.30 (0.74–2.29)	0.357	1.59 (0.79–3.17)	0.186
More than 10000 SAR	106	73	33	0.80 (0.45–1.44)	0.46	1.11 (0.47–2.60)	0.807
Smoking status							
Yes	63	35	28	Ref		Ref	
No	247	162	85	1.35 (0.94–1.79)	0.068	1.56 (1.17–2.41)	0.016
Duration of T2DM	9.84 ± 5.23			2.19 (0.69–4.34)	0.699	1.93 (0.65–4.31)	0.742
Type (s) of anti-diabetes treatment							
Insulin	82	47	35	Ref		Ref	
OHA	204	134	70	2.05 (0.69–4.19)	0.517	1.95 (0.83–3.14)	0.091
Both	24	16	8	0.89 (0.45–3.77)	0.864	1.22 (0.65–2.28)	0.112
Self-management behavior							
Satisfactory	98	66	32	Ref		Ref	
Unsatisfactory	212	131	81	1.73 (1.03–3.17)	0.008	1.55 (1.11–2.87)	0.027
Body mass index							
Normal	34	27	7	Ref		Ref	
Overweight	110	71	32	1.64 (0.81–2.67)	0.073	1.08 (0.71–1.96)	0.063
Obesity (all categories)	166	109	74	2.03 (1.21–3.71)		2.98 (1.31–5.07)	0.003
Presence of other chronic conditions							
No	208	130	78	Ref		Ref	
Yes	102	67	35	2.21 (1.35–3.60)	0.002	1.74 (1.11–2.67)	0.006

* Binomial logistic regression analysis (enter method) ** *p*-value < 0.05 is statistically significant value.

## Data Availability

The data presented in this study are available on request from the corresponding author. The data are not publicly available due to ethical reasons.

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
