# Peer review of "Evaluation of Self-Management Behaviors and Its Correlation with the Metabolic Syndrome among the Type 2 Diabetes Mellitus Patients of Northern Saudi Arabia"

_jcm, 2023, doi:10.3390/jcm13010118_

Round 1

Reviewer 1 Report

Comments and Suggestions for Authors

The paper titled "Evaluation of Self-Management Behaviors and Its Correlation with the Metabolic Syndrome among the Type 2 Diabetes Mellitus Patients of Northern Saudi Arabia" explores the prevalence of metabolic syndrome (MS) and its association with self-management behaviors in patients with type 2 diabetes mellitus (T2DM) in northern Saudi Arabia. The study involved 310 T2DM patients, assessing their self-management behaviors using the Summary of Diabetes Self-care Activities Scale and determining MS prevalence based on the National Cholesterol Education Program (NCEP) Adult Treatment Plan – 3 guidelines. The study suggests the need for improved self-management interventions to reduce MS and its complications in T2DM patients. It also acknowledges limitations, such as its cross-sectional design and the specific patient demographic. This research is significant for healthcare providers and policymakers, highlighting the importance of patient self-management in managing T2DM and its associated risks. Following are some comments for inference.

Comments:

1.     Scope and Relevance: The paper is highly relevant, as it addresses the crucial issue of self-management behaviors in diabetes care, which is a growing concern worldwide. However, it could benefit from delving deeper into how these self-management behaviors directly impact clinical outcomes in patients with Type 2 Diabetes Mellitus. This would provide greater insight into the practical applications of the study's findings.

2.     Figures and Data Interpretation: While the paper effectively utilizes statistical data to demonstrate its findings, the presentation of this data could be enhanced for clarity. For example, clearer visualization of the correlations between different self-management behaviors and metabolic syndrome components would aid in better understanding the study's implications.

3.     Discussion: The discussion provides a comprehensive overview of the findings and their significance. However, a more streamlined and focused approach could improve the paper's impact and readability. Reducing verbosity and focusing on key findings would make the conclusions more accessible to a broader audience.

4.     Clinical Implications: The paper outlines the correlation between self-management behaviors and metabolic syndrome, but it could further explore the direct clinical implications of these findings. Specifically, how can healthcare providers utilize this information to improve patient care and outcomes? Addressing this could greatly enhance the paper's practical value.

5.     Recommendations for Future Research: A section dedicated to future research recommendations would be a valuable addition. Identifying gaps in the current literature and suggesting areas for further investigation, especially in diverse demographic settings, would enhance the study's contribution to the field.

General Comments:

1.     Language and Clarity: The manuscript is well-written but could benefit from further editing for language and clarity. Some sections are somewhat dense, and simplifying the language could make the paper more accessible to non-specialist readers.

2.     Relevance to Human Models: The study focuses on a specific demographic, which is a strength in terms of detailed analysis. However, discussing how these findings might be extrapolated to different human populations, considering variations in lifestyle, healthcare systems, and cultural aspects, would add depth to the study's relevance.

Reviewer 2 Report

Comments and Suggestions for Authors

The study is well-designed. 

Moderate exercise encompasses both resistance and cardio training. Both have different effects on the cardiometabolic system. It is advised to include them separately (if possible).

The introduction is well-written. Methods and results are accurately described.

Units of BMI, waist circumference, and BP must be included in the tables as well as text.

Reviewer 3 Report

Comments and Suggestions for Authors

The title of the manuscript is interesting, unfortunately a series of improvements are needed to achieve the goal set in the title:

- food and lifestyle management is very important in type 2 diabetes, as a result a correlation between physical activity and the severity of imbalances in the patient group is very important, most are overweight or obese, are they sedentary?, what is the level of physical activity by BMI category, severity of dyslipidemia, etc.

- correlation between the amount of vegetable products consumed (intake of fibers, vitamins, minerals, antioxidants) and BMI, serum lipids, possibly blood sugar;

- presentation of the limits of the study and its benefits;

- improving the conclusions

Round 2

Reviewer 3 Report

Comments and Suggestions for Authors

Pay attention to the numbering of the tables, table 4 appears twice, the last one is table 5 and must be corrected in the explanatory text.